# GENERATIVE NETWORKS AS INVERSE PROBLEMS WITH SCATTERING TRANSFORMS

**Tomás Angles & Stéphane Mallat**
École normale supérieure, Collège de France, PSL Research University
75005 Paris, France
`tomas.angles@ens.fr`

## ABSTRACT

Generative Adversarial Nets (GANs) and Variational Auto-Encoders (VAEs) provide impressive image generations from Gaussian white noise, but the underlying mathematics are not well understood. We compute deep convolutional network generators by inverting a fixed embedding operator. Therefore, they do not require to be optimized with a discriminator or an encoder. The embedding is Lipschitz continuous to deformations so that generators transform linear interpolations between input white noise vectors into deformations between output images. This embedding is computed with a wavelet Scattering transform. Numerical experiments demonstrate that the resulting Scattering generators have similar properties as GANs or VAEs, without learning a discriminative network or an encoder.

## 1 INTRODUCTION

Generative Adversarial Networks (GANs) and Variational Auto-Encoders (VAEs) allow training generative networks to synthesize images of remarkable quality and complexity from Gaussian white noise. This work shows that one can train generative networks having similar properties to those obtained with GANs or VAEs without learning a discriminator or an encoder. The generator is a deep convolutional network that inverts a predefined embedding operator. To reproduce relevant properties of GAN image synthesis the embedding operator is chosen to be Lipschitz continuous to deformations, and it is implemented with a wavelet Scattering transform. Defining image generators as the solution of an inverse problem provides a mathematical framework, which is closer to standard probabilistic models such as Gaussian autoregressive models.

GANs were introduced by Goodfellow et al. (2014) as an unsupervised learning framework to estimate implicit generative models of complex data (such as natural images) by training a generative model (the generator) and a discriminative model (the discriminator) simultaneously. An implicit generative model of the random vector $X$ consists in an operator $\widehat{G}$ which transforms a Gaussian white noise random vector $Z$ into a model $\widehat{X} = \widehat{G}(Z)$ of $X$. The operator $\widehat{G}$ is called a generative network or generator when it is a deep convolutional network. Radford et al. (2016) introduced deep convolutional architectures for the generator and the discriminator, which result in high-quality image synthesis. They also showed that linearly modifying the vector $z$ results in a progressive deformation of the image $\hat{x} = \widehat{G}(z)$.

Goodfellow et al. (2014) and Arjovsky et al. (2017) argue that GANs select the generator $\widehat{G}$ by minimizing the Jensen-Shannon divergence or the Wasserstein distance calculated from empirical estimations of these distances with generated and training images. However, Arora et al. (2017) prove that this explanation fails to pass the curse of dimensionality since estimates of Jensen-Shannon or Wasserstein distances do not generalize with a number of training examples which is polynomial on the dimension of the images. Therefore, the reason behind the generalization capacities of generative networks remains an open problem.

VAEs, introduced by Kingma & Welling (2014), provide an alternative approach to GANs, by optimizing $\widehat{G}$ together with its inverse on the training samples, instead of using a discriminator. The inverse $\Phi$ is an embedding operator (the encoder) that is trained to transform $X$ into a Gaussian white noise $Z$. Therefore, the loss function to train a VAE is based on probabilistic distances which

also suffer from the same dimensionality curse shown in Arora et al. (2017). Furthermore, a significant disadvantage of VAEs is that the resulting generative models produce blurred images compared with GANs.

Generative Latent Optimization (GLO) was introduced in Bojanowski et al. (2017) to eliminate the need for a GAN discriminator while restoring sharper images than VAEs. GLO still uses an autoencoder computational structure, where the latent space variables $z$ are optimized together with the generator $G$. Despite good results, linear variations of the embedding space variables are not mapped as clearly into image deformations as in GANs, which reduces the quality of generated images.

GANs and VAEs raise many questions. Where are the deformation properties coming from? What are the characteristics of the embedding operator $\Phi$? Why do these algorithms seem to generalize despite the curse of dimensionality? Learning a stable embedding which maps $X$ into a Gaussian white noise is intractable without strong prior information (Arora et al., 2017). This paper shows that this prior information is available for image generation and that one can predefine the embedding up to a linear operator. The embedding must be Lipschitz continuous to translations and deformations so that modifications of the input noise result in deformations of $\widehat{X}$. Lipschitz continuity to deformations requires separating the signal variations at different scales, which leads to the use of wavelet transforms. We concentrate on wavelet Scattering transforms (Mallat, 2012), which linearize translations and provide appropriate Gaussianization. We then define the generative model as an inversion of the Scattering embedding on training data, with a deep convolutional network. The inversion is regularized by the architecture of the generative network, which is the same as the generator of a DCGAN (Radford et al., 2016). Experiments in Section 4 show that these generative Scattering networks have similar properties as GAN generators, and the synthesized images have the same quality as the ones obtained with VAEs or GLOs.

## 2 COMPUTING A GENERATOR FROM AN EMBEDDING

### 2.1 GENERATOR CALCULATION AS AN INVERSE PROBLEM

Unsupervised learning consists in estimating a model $\widehat{X}$ of a random vector $X$ of dimension $p$ from $n$ realizations $\{x_i\}_{i \leq n}$ of $X$. Autoregressive Gaussian processes are simple models $\widehat{X} = \widehat{G}(Z)$ computed from an input Gaussian white noise $Z$ by estimating a parametrized linear operator $\widehat{G}$. This operator is obtained by inverting a linear operator, whose coefficients are calculated from the realizations $\{x_i\}_{i \leq n}$ of $X$. We shall similarly compute models $\widehat{X} = \widehat{G}(Z)$ from a Gaussian white noise $Z$, by estimating a parametrized operator $\widehat{G}$, but which is a deep convolutional network instead of a linear operator. $\widehat{G}$ is obtained by inverting an embedding $\{\Phi(x_i)\}_{i \leq n}$ of the realizations $\{x_i\}_{i \leq n}$ of $X$, with a predefined operator $\Phi(x)$.

Let us denote by $\mathcal{G}$ the set of all parametrized convolutional network generators defined by a particular architecture. We impose that $\widehat{G}(\Phi(x_i)) \approx x_i$ by minimizing an $L^1$ loss over the convolutional network class $\mathcal{G}$:

$$\widehat{G} = \underset{G \in \mathcal{G}}{\operatorname{argmin}} \, n^{-1} \sum_{i=1}^{n} \|x_i - G(\Phi(x_i))\|_1 . \tag{1}$$

We use the $L^1$ norm because it has been reported (e.g., Bojanowski et al. (2017)) to give better results for natural signals such as images. The resulting generator $\widehat{G}$ depends upon the training examples $\{x_i\}_{i \leq n}$ and on the regularization imposed by the network class $\mathcal{G}$. We shall say that the network generalizes over $\Phi(X)$ if $\mathbb{E}(\|X - \widehat{G}(\Phi(X))\|_1)$ is small and comparable to the empirical error $n^{-1} \sum_{i=1}^{n} \|x_i - \widehat{G}(\Phi(x_i))\|_1$. We say that the network generalizes over the Gaussian white noise $Z$ if realizations of $\widehat{X} = \widehat{G}(Z)$ are realizations of $X$. If $\widehat{G}$ generalizes over $\Phi(X)$, then a sufficient condition to generalize over $Z$ is that $\Phi$ transforms $X$ into Gaussian white noise, i.e. $\Phi$ gaussianizes $X$. Besides this condition, the role of the embedding $\Phi$ is to specify the properties of $\hat{x} = \widehat{G}(z)$ that should result from modifications of $z$.

We shall define $\Phi = A\overline{\Phi}$, where $\overline{\Phi}$ is a fixed normalized operator, and $A$ is an affine operator which performs a whitening and a projection to a lower dimensional space. We impose that $\{\overline{\Phi}(x_i)\}_{i \leq n}$ are realizations of a Gaussian process and that $A$ transforms this process into a lower dimensional Gaussian white noise. We normalize $\overline{\Phi}$ by imposing that $\overline{\Phi}(0) = 0$ and that it is contractive, this is:

$$\forall (x, x') \in \mathbb{R}^{2p} \ , \ \|\overline{\Phi}(x) - \overline{\Phi}(x')\| \leq \|x - x'\| \ .$$

The affine operator $A$ performs a whitening of the distribution of the $\{\overline{\Phi}(x_i)\}_{i \leq n}$ by subtracting the empirical mean $\mu$ and normalizing the largest eigenvalues of the empirical covariance matrix $\Sigma$. For this purpose, we calculate the eigendecomposition of the covariance matrix $\Sigma = Q D Q^T$. Let $P_{V_d} = Q_d Q_d^T$ be the orthogonal projection in the space generated by the $d$ eigenvectors having the largest eigenvalues. We choose $d$ so that $P_{V_d}\overline{\Phi}(x)$ does not contract too much the distances between the $\{x_i\}_{i \leq n}$, formally, this means that $P_{V_d}\overline{\Phi}(x)$ defines a bi-Lipschitz embedding of these samples, and hence that there exists a constant $\alpha > 0$ so that:

$$\forall i, i' \leq n \ , \ \frac{1}{\alpha}\|x_i - x_{i'}\| \leq \|P_{V_d}\overline{\Phi}(x_i) - P_{V_d}\overline{\Phi}(x_{i'})\| \leq \|x_i - x_{i'}\| \ . \tag{2}$$

This bi-Lipschitz property must be satisfied when $d$ is equal to the dimension of $\overline{\Phi}$ and hence when $P_{V_d}$ is the identity. The dimension $d$ should then be adjusted to avoid reducing too much the distances between the $\{x_i\}_{i \leq n}$.

We choose $A = \Sigma_d^{-1/2}(Id - \mu)$ with $\Sigma_d^{-1/2} = D_d^{-1/2}Q_d^T$, where $Id$ is the identity. The resulting embedding is:

$$\Phi(x) = \Sigma_d^{-1/2}(\overline{\Phi}(x) - \mu) \ .$$

This embedding computes a $d$-dimensional whitening of the $\{\overline{\Phi}(x_i)\}_{i \leq n}$. The generator network $\widehat{G}$ which inverts $\Phi$ over the training samples is then computed according to (1).

**Associative Memory:** The existence of a known embedding operator $\Phi$ allows us to use the generative network $\widehat{G}$ as an associative or content addressable memory. The input $Z$ can be interpreted as an address, of lower dimension $d$ than the generated image $\widehat{X}$. Any training image $x_i$ is associated to the address $z_i = \Phi(x_i)$. The network is optimized to generate the training images $\{x_i\}_{i \leq n}$ from these lower dimensional addresses. The inner network coefficients thus include a form of distributed memory of these training images. Moreover, if the network generalizes then one can approximately reconstruct a realization $x$ of the random process $X$ from its embedding address $z = \Phi(x)$. In this sense, the memory is content addressable.

## 2.2 Gaussianization and Continuity to Deformations

We now describe the properties of the normalized embedding operator $\overline{\Phi}$ to build a generator having similar properties as GANs or VAEs. We mentioned that we would like $\Phi(X)$ to be nearly a Gaussian white noise and since $\Phi = A\overline{\Phi}$ where $A$ is affine then $\overline{\Phi}(X)$ should be Gaussian. Therefore, $\{\overline{\Phi}(x_i)\}_{i \leq n}$ should be realizations of a Gaussian process and hence be concentrated over an ellipsoid.

The normalized embedding operator $\overline{\Phi}$ must also be covariant to translations because it will be inverted by several layers of a deep convolutional generator, which are covariant to translations. Indeed, the generator belongs to $\mathcal{G}$ which is defined by a DCGAN architecture (Radford et al., 2016). In this family of networks, the first layer is a linear operator which reshapes and adjusts the mean and covariance of the input white noise $Z$. The non-stationary part of the process $\widehat{X} = \widehat{G}(Z)$ is captured by this first affine transformation. The next layers are all convolutional and hence covariant to translations. These layers essentially invert the normalized embedding operator $\overline{\Phi}$ over the training samples.

The normalized embedding operator $\overline{\Phi}$ must also linearize translations and small deformations. Indeed, if the input $z = A\overline{\Phi}(x)$ is linearly modified then, to reproduce GAN properties, the output

$\widehat{G}(z)$ should be continuously deformed. Therefore, we require $\overline{\Phi}$ to be Lipschitz continuous to translations and deformations. A translation and a deformation of an image $x(u)$ can be written as $x_\tau(u) = x(u - \tau(u))$, where $u$ denotes the spatial coordinates. Let $|\tau|_\infty = \max_u |\tau(u)|$ be the maximum translation amplitude. Let $\nabla\tau(u)$ be the Jacobian of $\tau$ at $u$ and $|\nabla\tau(u)|$ be the norm of this Jacobian matrix. The deformation induced by $\tau$ is measured by $|\nabla\tau|_\infty = \max_u |\nabla\tau(u)|$, which specifies the maximum scaling factor induced by the deformation. The value $|\nabla\tau|_\infty$ defines a metric on diffeomorphisms (Mallat, 2012) and thus specifies the deformation "size". The operator $\overline{\Phi}$ is said to be Lipschitz continuous to translations and deformations over domains of scale $2^J$ if there exists a constant $C$ such that for all $x$ and all $\tau$ we have:

$$\|\overline{\Phi}(x) - \overline{\Phi}(x_\tau)\| \leq C \|x\| \left( 2^{-J} |\tau|_\infty + |\nabla\tau|_\infty \right). \tag{3}$$

This inequality implies that translations of $x$ that are small relative to $2^J$ and small deformations are mapped by $\overline{\Phi}$ into small quasi-linear variations of $z = A\overline{\Phi}(x)$.

The Gaussianization property means that $\{\overline{\Phi}(x_i)\}_{i \leq n}$ should be concentrated on an ellipsoid. This is always true if $n \leq p$, but it can be difficult to achieve if $n \gg p$. In one dimension $x \in \mathbb{R}$, an invertible differentiable operator $\Phi$ which Gaussianizes a random variable $X$ can be computed as the solution of a differential equation that transports the histogram into a Gaussian (Friedman, 1987). In higher dimensions, this strategy has been extended by iteratively performing a Gaussianization of one-dimensional variables, through independent component analysis (Chen & Gopinath, 2000) or with random rotations (Laparra et al., 2011). However, these approaches do not apply in this context because they do not necessarily define operators which are translation covariant and Lipschitz continuous to deformations.

Another Gaussianization strategy comes from the Central Limit Theorem by averaging nearly independent random variables having variances of the same order of magnitude. This averaging can be covariant to translations if implemented with convolutions with a low-pass filter. The resulting operator will also be Lipschitz continuous to deformations. However, an averaging operator loses all high-frequency information. To define an operator $\overline{\Phi}$ which satisfies the bi-Lipschitz condition (2), we must preserve the high-frequency content despite the averaging. The next section explains how to do so with a Scattering transform.

## 3 GENERATIVE SCATTERING NETWORKS

In this section, we show that a Scattering transform (Mallat, 2012; Bruna & Mallat, 2013) provides an appropriate embedding for image generation, without learning. It does so by taking advantage of prior information on natural signals, such as translation and deformation properties. We also specify the architecture of the deep convolutional generator that inverts this embedding, and we summarize the algorithm to perform this regularized inversion.

Since the first order term of a deformation is a local scaling, defining an operator that is Lipschitz continuous to deformations, and hence satisfies (3), requires decomposing the signal at different scales, which is done by a wavelet transform (Mallat, 2012). The linearization of translations at a scale $2^J$ is obtained by an averaging implemented with a convolution with a low-pass filter at this scale. A Scattering transform uses non-linearities to compute interactions between coefficients across multiple scales, which restores some information lost due to the averaging. It also defines a bi-Lipschitz embedding (2) and the averaging at the scale $2^J$ Gaussianizes the random vector $X$. This scale $2^J$ adjusts the trade-off between Gaussianization and contraction of distances due to the averaging.

A Scattering operator $S_J$ transforms $x(u)$ into a tensor $x_J(u, k)$, where the spatial parameter $u$ is sampled at intervals of size $2^J$ and the channels are indexed by $k$. The number $K_J$ of channels increases with $J$ to partially compensate for the loss of spatial resolution. These $K_J$ channels are computed by a non-linear translation covariant transformation $\Psi_J$. In this paper, $\Psi_J$ is computed as successive convolutions with complex two-dimensional wavelets followed by a pointwise complex modulus, with no channel interactions. Following Bruna & Mallat (2013), we choose a Morlet wavelet $\psi$, scaled by $2^\ell$ for different values of $\ell$ and rotated along $Q$ angles $\theta = q\pi/Q$:

$$\psi_{\ell,q}(u) = 2^{-2\ell}\psi(2^{-\ell}r_\theta u) \text{ for } 0 \le q < Q .$$

To obtain an order two Scattering operator $S_J$, the operator $\Psi_J$ computes sequences of up to two wavelet convolutions and complex modulus:

$$\Psi_J(x) = \Big[ x \, , \, |x \star \psi_{\ell,q}| \, , \, ||x \star \psi_{\ell,q}| \star \psi_{\ell',q'}| \Big]_{1 \le \ell < \ell' \le J, \, 1 \le q, q' \le Q} .$$

Therefore, there are $K_J = 1 + QJ + Q^2J(J-1)/2$ channels. A Scattering transform is then obtained by averaging each channel with a Gaussian low-pass filter $\phi_J(u) = 2^{-2J}\phi(2^{-J}u)$ whose spatial width is proportional to $2^J$:

$$S_J(x) = \Psi_J(x) \star \phi_J = \Big[ x \star \phi_J \, , \, |x \star \psi_{\ell,q}| \star \phi_J \, , \, ||x \star \psi_{\ell,q}| \star \psi_{\ell',q'}| \star \phi_J \Big]_{1 \le \ell < \ell' \le J, \, 1 \le q, q' \le Q} .$$

Convolutions with $\phi_J$ are followed by a subsampling of $2^J$; as a result, if $x$ has $p$ pixels then $S_J$ is of dimension $p\,\alpha_J$ where:

$$\alpha_J = 2^{-2J}(1 + QJ + Q^2J(J-1)/2) .$$

The maximum scale $2^J$ is limited by the image width $2^J \le p^{1/2}$. For our experiments we used $Q = 8$ and images $x$ of size $p = 128^2$. In this case $\alpha_4 = 1.63$ and $\alpha_5 = 0.67$. Since $\alpha_J > 1$ for $J \le 4$, $S_J(x)$ has more coefficients than $x$. Based on this coefficient counting, we expect $S_J$ to be invertible for $J \le 4$, but not for $J \ge 5$.

The wavelets separate the variations of $x$ at different scales $2^\ell$ along different directions $q\pi/Q$, and second order Scattering coefficients compute interactions across scales. Because of this scale separation, one can prove that $S_J$ is Lipschitz continuous to translations and deformations (Mallat, 2012) in the sense of eq. (3). Wavelets also satisfy a Littlewood-Paley condition which guarantees that the wavelet transform and also $S_J$ are contractive operators (Mallat, 2012).

If wavelet coefficients become nearly independent when they are sufficiently far away then $S_J(X)$ becomes progressively more Gaussian as the scale $2^J$ increases, because of the spatial averaging by $\phi_J$. Indeed, if $X(u)$ is independent from $X(v)$ for $|x - v| \ge \Delta$ then the Central Limit Theorem proves that $S_J(X)$ converges to a Gaussian distribution when $2^J/\Delta$ increases. However, as the scale increases, the averaging produces progressively more instabilities which can deteriorate the bi-Lipschitz bounds in (2). This trade-off between Gaussianization and stability defines the optimal choice of the scale $2^J$.

As explained in (Mallat, 2016), convolutions with wavelets $\psi_{\ell,q}$ and the low-pass filter $\phi_J$ can also be implemented as convolutions with small size filters and subsamplings. As a result, a Scattering transform is obtained by cascading convolution matrices $V_j$ and the complex modulus as a non-linearity:

$$S_j = |V_j S_{j-1}| \text{ for } 1 \le j \le J .$$

A Scattering transform is thus an instance of a deep convolutional network whose filters are specified by wavelets and where the non-linearity is chosen to be a modulus. The choice of filters is flexible and other multiscale representations, such as the ones in Portilla & Simoncelli (2000); Lyu & Simoncelli (2009); Malo & Laparra (2010), may also be used.

Following the notations in 2.1, the normalized embedding operator $\overline{\Phi}$ is chosen to be $S_J$, thus the embedding operator is defined by $\Phi(x) = \Sigma_d^{-1/2}(S_J(x) - \mu)$. A generative Scattering network is a deep convolutional network which implements a regularized inversion of this embedding. Both networks are illustrated in Figure 1.

More specifically, a generative Scattering network is a deep convolutional network $\widehat{G}$ which inverts the whitened Scattering embedding $\Phi$ on training samples. It is obtained by minimizing the $L^1$ loss

$$X \rightarrow \boxed{|V_1|} \rightarrow \boxed{|V_2|} \rightarrow \cdots \rightarrow \boxed{|V_J|} \rightarrow S_J(X) \rightarrow \boxed{\Sigma_d^{-1/2}(Id - \mu)} \rightarrow \Phi(X)$$

$$Z \rightarrow \boxed{\rho W_0} \rightarrow \boxed{\rho W_1} \rightarrow \cdots \rightarrow \boxed{\rho W_{J+1}} \rightarrow \widehat{X}$$

Figure 1: Top: the embedding operator $\Phi$ consists of a Scattering transform $S_J(X)$ computed by cascading $J$ convolutional wavelet operators $V_j$ followed by a pointwise complex modulus and then an affine whitening. Bottom: The generative network is computed by cascading Relus $\rho$ and linear operators plus biases $W_j$, which are convolutional along spatial variables for $j \geq 1$.

$n^{-1} \sum_{i=1}^{n} \|G(\Phi(x_i)) - x_i\|_1$, as explained in Section 2.1. The minimization is done with the Adam optimizer (Kingma & Ba, 2014), using the default hyperparameters. The generator illustrated in Figure 1, is a DCGAN generator (Radford et al., 2016), of depth $J + 2$:

$$G = \rho\, W_{J+1}\, \rho\, W_J \,...\, \rho\, W_1\, \rho\, W_0\, .$$

The non-linearity $\rho$ is a ReLU. The first operator $W_0$ is linear (fully-connected) plus a bias, it transforms $Z$ into a $4 \times 4$ array of 1024 channels. The next operators $W_j$ for $1 \leq j \leq J$ perform a bilinear upsampling of their input, followed by a multichannel convolution along the spatial variables, and the addition of a constant bias for each channel. The operators $\rho W_j$ compute a progressive inversion of $S_J(x)$, calculated with the convolutional operators $|V_j|$ for $1 \leq j \leq J$. The last operator $W_{J+1}$ does not perform an upsampling. All the convolutional layers have filters of size 7, with symmetric padding at the boundaries. All experiments are performed with color images of dimension $p = 128^2 \times 3$ pixels.

## 4 NUMERICAL EXPERIMENTS

This section evaluates generative Scattering networks with several experiments. The accuracy of the inversion given by the generative network is first computed by calculating the reconstruction error of training images. We assess the generalization capabilities by computing the reconstruction error on test images. Then, we evaluate the visual quality of images generated by sampling the Gaussian white noise $Z$. Finally, we verify that linear interpolations of the embedding variable $z = \Phi(x)$ produce a morphing of the generated images, as in GANs. The code to reproduce the experiments can be found in [1].

We consider three datasets that have different levels of variabilities: CelebA (Liu et al., 2015), LSUN (bedrooms) (Yu et al., 2015) and Polygon5. The last dataset consists of images of random convex polygons of at most five vertices, with random colors. All datasets consist of RGB color images with shape $128^2 \times 3$. For each dataset, we consider only 65536 training images and 16384 test images.

In all experiments, the Scattering averaging scale is $2^J = 2^4 = 16$, which linearizes translations and deformations of up to 16 pixels. For an RGB image $x$, $S_4(x)$ is computed for each of the three color channels. Because of the subsampling by $2^{-4}$, $S_4(x)$ has a spatial resolution of $8 \times 8$, with $417 \times 3 = 1251$ channels, and is thus of dimension $\approx 8 \times 10^4$. Since it has more coefficients than the input image $x$, we expect it to be invertible, and numerical experiments indicate that this is the case. This dimension is reduced to $d = 512$ by the whitening operator. The resulting operator remains a bi-Lipschitz embedding of the training images of the three datasets in the sense of (2). The Lipschitz constant is $\alpha = 5$, and $99.5\%$ of the distances between image pairs $(x_i, x_{i'})$ are preserved with a Lipschitz constant smaller than 3. Further reducing the dimension $d$ to 100, which is often used in numerical experiments (Radford et al., 2016; Bojanowski et al., 2017), has a marginal effect on the Lipschitz bound and on numerical results, but it slightly affects the recovery of high-frequency details.

We now assess the generalization properties of $\widehat{G}$ by comparing the reconstructions of training and test samples from $\Phi(X) = \Sigma_d^{-1/2}(S_4(X) - \mu)$; figures 2 and 3 show such reconstructions. Table 1

---

[1]https://github.com/tomas-angles/generative-scattering-networks

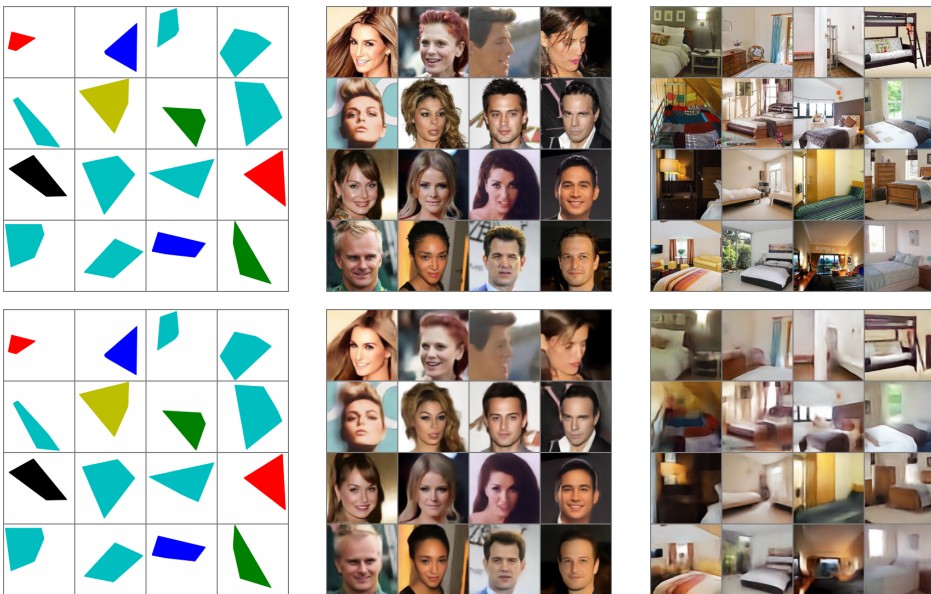

Figure 2: Top: original images $x$ in the training set. Bottom: reconstructions from $\Phi(x)$ using $\widehat{G}$.

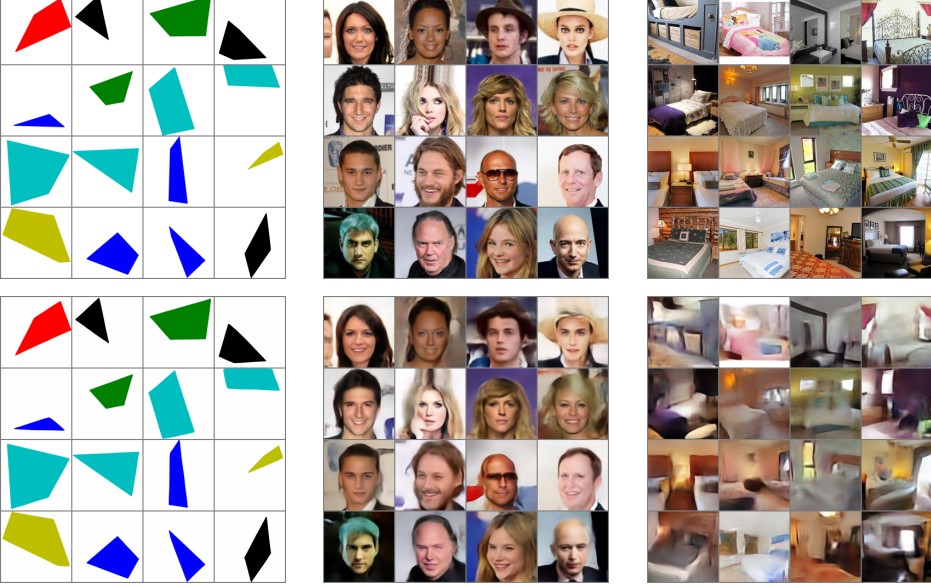

Figure 3: Top: original images $x$ in the test set. Bottom: reconstructions from $\Phi(x)$ using $\widehat{G}$.

gives the average training and test reconstruction errors in dB for each dataset. The training error is between 3dB and 8dB above the test error, which is a sign of overfitting. However, this overfitting is not large compared to the variability of errors from one dataset to the next. Overfitting is not good for unsupervised learning where the intent is to model a probability density, but if we consider this network as an associative memory, it is not a bad property. Indeed, it means that the network performs a better recovery of known images used in training than unknown images in the test, which is needed for high precision storage of particular images.

Polygons are simple images that are much better recovered than faces in CelebA, which are simpler images than the ones in LSUN. This simplicity is related to the sparsity of their wavelet coefficients, which is higher for polygons than for faces or bedrooms. Wavelet sparsity drives the properties of Scattering coefficients which provide localized $l^1$ norms of wavelet coefficients. The network reg-

ularizes the inversion by storing the information needed to reconstruct the training images, which is a form of memorization. LSUN images require more memory because their wavelet coefficients are less sparse than polygons or faces; this might explain the difference of accuracies over datasets. However, the link between sparsity and the memory capacity of the network is not yet fully understood. The generative network has itself sparse activations with about $70\%$ of them being equal to zero on average over all images of the three datasets. Sparsity thus seems to be an essential regularization property of the network.

|  | CelebA | Polygon5 | LSUN (bedrooms) |
|---|---|---|---|
| **Train** | 25.95 | 42.43 | 21.77 |
| **Test** | 21.17 | 34.44 | 18.53 |

Table 1: PSNR reconstruction errors in dB of train and test images, from their whitened Scattering embedding, over three datasets.

Similarly to VAEs and GLOs, Scattering image generations eliminate high-frequency details, even on training samples. This is due to a lack of memory capacity of the generator. This was verified by reducing the number of training images. Indeed, when using only $n = 256$ training images, all high-frequencies are recovered, but there are not enough images to generalize well on test samples. This is different from GAN generations, where we do not observe this attenuation of high-frequencies over generated images. GANs seem to use a different strategy to cope with the memory limitation; instead of reducing precision, they seem to "forget" some training samples (mode-dropping), as shown in Bojanowski et al. (2017). Therefore, GANs versus VAEs or generative Scattering networks introduce different types of errors, which affect diversity versus recovery precision.

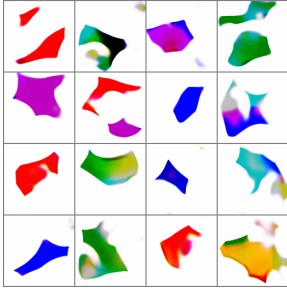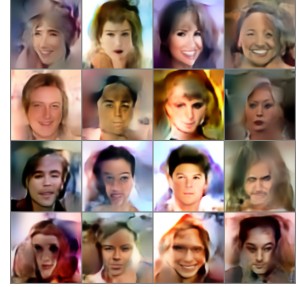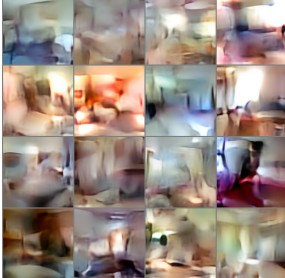

Figure 4: Images $\widehat{X} = \widehat{G}(Z)$ generated from a Gaussian white noise $Z$.

To evaluate the generalization properties of the network on Gaussian white noise $Z$, Figure 4 shows images $\widehat{X} = \widehat{G}(Z)$ generated from random samplings of $Z$. Generated images have strong similarities with the ones in the training set for polygons and faces. The network recovers colored geometric shapes in the case of Polygon5 even tough they are not exactly polygons, and it recovers faces for CelebA with a blurred background. For LSUN, the images are piecewise regular, and most high frequencies are missing; this is due to the complexity of the dataset and the lack of memory capacity of the generative network.

Figure 5 evaluates the deformation properties of the network. Given two input images $x$ and $x'$, we modify $\alpha \in [0, 1]$ to obtain the interpolated images:

$$x_\alpha = \widehat{G}\Big((1 - \alpha)z + \alpha z'\Big) \text{ for } z = \Phi(x) \text{ and } z' = \Phi(x') . \tag{4}$$

The linear interpolation over the embedding variable $z$ produces a continuous deformation from one image to the other while colors and image intensities are also adjusted. It reproduces the morphing properties of GANs. In our case, these properties result from the Lipschitz continuity to deformations of the Scattering transform.

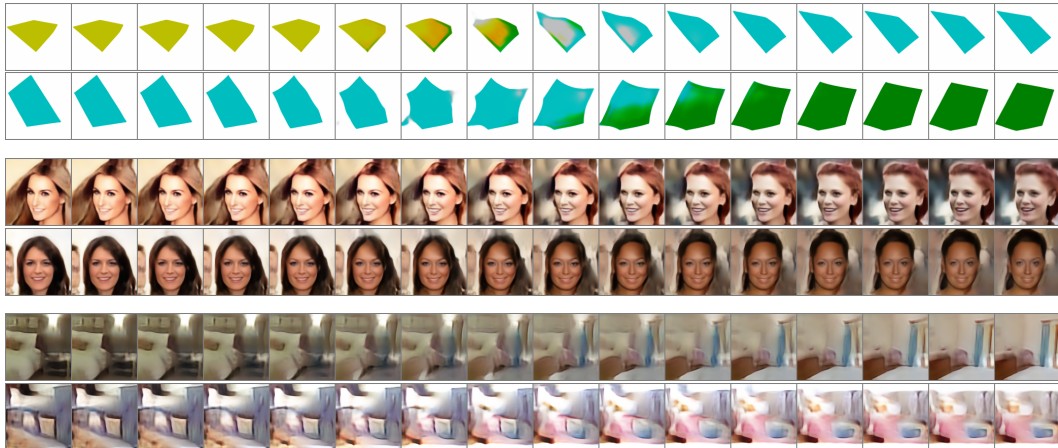

Figure 5: Each row shows the image morphing obtained by linearly interpolating the latent variables of the left-most and right-most images, according to (4). For each of the three datasets, the first row is computed with two training images and the second row with two test images.

## 5 CONCLUSION

This paper shows that most properties of GANs and VAEs can be reproduced with an embedding computed with a Scattering transform, which avoids using a discriminator as in GANs or learning the embedding as in VAEs or GLOs. It also provides a mathematical framework to analyze the statistical properties of these generators through the resolution of an inverse problem, regularized by the convolutional network architecture and the sparsity of the obtained activations. Because the embedding function is known, numerical results can be evaluated on training as well as test samples.

We report preliminary numerical results with no hyperparameter optimization. The architecture of the convolutional generator may be adapted to the properties of the Scattering operator $S_j$ as $j$ increases. Also, the paper uses a "plain" Scattering transform which does not take into account interactions between angle and scale variables, which may also improve the representation as explained in Oyallon & Mallat (2015).

## ACKNOWLEDGEMENTS

This work was funded by the ERC grant InvariantClass 320959.

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
