# OpenReview forum: "Generative networks as inverse problems with Scattering transforms"
_ICLR.cc/2018/Conference — Accept (Poster)_

### Official Review · AnonReviewer1 · 2017-11-26
**interesting and original insights on VAE/GAN with the scattering transform**

**Rating:** 7
**Confidence:** 4

**Review:**

After a first manuscript that needed majors edits, the revised version
offers an interesting GAN approach based the scattering transform.

Approach is well motivated with proper references to the recent literature.

Experiments are not state of the art but clearly demonstrate that the
proposed approach does provide meaningful results.

---

> ### Author Response · Authors · 2017-12-28
> **Answers to AnonReviewer1**
>
> Remark: Please do not miss the thank you comment (above) for all the reviews.
>
> We rewrote large parts of the paper to make it as clear as possible. Hopefully, as expressed in the thank you comment, the changes we made will make things clearer.

---

### Official Review · AnonReviewer2 · 2017-11-27
**Authors generate images from Gaussian noise using scattering networks. Interesting analysis of Gaussianization transforms constrained to be continuous to deformations -> Accept**

**Rating:** 8
**Confidence:** 4

**Review:**

The authors introduce scattering transforms as image generative models in the context of Generative Adversarial Networks and suggest why they could be seen as Gaussianization transforms with controlled information loss and invertibility.
Writing is suggestive and experimental results are interesting, so I clearly recommend acceptation.

I would appreciate more intuition on some claims (e.g. relation between Lipschitz continuity and wavelets) but they refer to the appropriate reference to Mallat, so this is not a major problem for the interested reader.

However, related to the above non-intuitive claim, here is a question on a related Gaussianization transform missed by the authors that (I feel) fulfils the conditions defined in the paper but it is not obviously related to wavelets. Authors cite Chen & Gopinath (2000) and critizise that their approach suffers from the curse of dimensionality because of the ICA stage. However, other people [Laparra et al. Iterative Gaussianization from ICA to random rotations IEEE Trans.Neural Nets 2011] proved that the ICA stage is not required (but only marginal operations followed by even random rotations). That transform seems to be Lipschitz continuous as well -since it is smooth and derivable-. In fact it has been also used for image synthesis. However, it is not obviously related to wavelets... Any comment?

Another relation to previous literature: in the end, the proposed analysis (or Gaussianization) transform is basically a wavelet transform where the different scale filters are applied in a cascade (fig 1). This is similar to Gaussian Scale Mixture  models for texture analysis [Portilla & Simoncelli Int. J. Comp. Vis. 2000] in which after wavelet transform, local division is performed to obtain Gaussian variables, and these can be used to synthesize the learned textures. That is similar to Divisive Normalization models of visual neuroscience that perform similar normalization alfter wavelets to factorize the PDF (e.g. [Lyu&Simoncelli Radial Gaussianization Neur.Comput. 2009], or [Malo et al. Neur.Comput. 2010]).

Minor notation issues: authors use a notation for functions that seems confusing (to me) since it looks like linear products. For instance: GZ for G(Z) [1st page] and phiX for phi(X) [2nd page] Sx for S(x) [in page 5]...

---

> ### Author Response · Authors · 2017-12-28
> **Answers to AnonReviewer2**
>
> Remark: Please do not miss the thank you comment (above) for all the reviews.
>
> Question: "I would appreciate more intuition on some claims (e.g. relation between Lipschitz continuity and wavelets) but they refer to the appropriate reference to Mallat, so this is not a major problem for the interested reader."
>
> We have included now the definition of Lipschitz continuity to deformations in order to specify more clearly what it means, and we give a short, intuitive explanation of what is required at the beginning of section 3.1. Going beyond would be too long, so we referred to the paper of Mallat (2012).
>
> Question: "However, related to the above non-intuitive claim, here is a question on a related Gaussianization transform missed by the authors that (I feel) fulfils the conditions defined in the paper but it is not obviously related to wavelets. Authors cite Chen \& Gopinath (2000) and critizise that their approach suffers from the curse of dimensionality because of the ICA stage. However, other people [Laparra et al. Iterative Gaussianization from ICA to random rotations IEEE Trans.Neural Nets 2011] proved that the ICA stage is not required (but only marginal operations followed by even random rotations). That transform seems to be Lipschitz continuous as well -since it is smooth and derivable-. In fact it has been also used for image synthesis. However, it is not obviously related to wavelets... Any comment?"
>
> These transforms are Lipschitz in the sense that a small additive modification of the input yields a small modification of the output. The Lipschitz continuity to deformations means that a small modification of the form of a dilation yields a small modification of the Euclidean norm of the resulting vector. However, a small dilation can induce a large displacement of high frequencies. To avoid this, it is necessary to separate the frequencies in different packets, which is done by wavelets, and map them back to lower frequencies, which is done by the modulus and averaging (a rectifier could replace the modulus). We now explain these points at the beginning of section 3.1.
>
> For this reason, it is very unlikely that an Iterative Gaussianization produces an operator that is stable to deformations unless they take into account this issue, but in [Laparra et al. IEEE Trans.Neural Nets 2011] there is no mention of this fact. While it is true that they also synthesize images, they do not show results on the nature of the interpolations between them.
>
> Question: "Another relation to previous literature: in the end, the proposed analysis (or Gaussianization) transform is basically a wavelet transform where the different scale filters are applied in a cascade (fig 1). This is similar to Gaussian Scale Mixture  models for texture analysis [Portilla \& Simoncelli Int. J. Comp. Vis. 2000] in which after wavelet transform, local division is performed to obtain Gaussian variables, and these can be used to synthesize the learned textures. That is similar to Divisive Normalization models of visual neuroscience that perform similar normalization alfter wavelets to factorize the PDF (e.g. [Lyu\&Simoncelli Radial Gaussianization Neur.Comput. 2009], or [Malo et al. Neur.Comput. 2010])."
>
> Yes, there are similarities between Portilla and Simoncelli representations and scattering representations, and we have included them in the references. Portilla and Simoncelli also use the modulus of wavelet coefficients. At the second order, they use a covariance operator; this may create a problem because covariance operators are not entirely stable to deformations (this is better explained in the paper of Mallat (2012)). One may indeed define embedding operators, different from the scattering transform, which could lead to as good and maybe better results. We now emphasize this critical point in the introduction and at the beginning of Section 3.1 and refer to all these papers.
>
> Question: "Minor notation issues: authors use a notation for functions that seems confusing (to me) since it looks like linear products. For instance: GZ for G(Z) [1st page] and phiX for phi(X) [2nd page] Sx for S(x) [in page 5]..."
>
> This was modified.

---

### Official Review · AnonReviewer3 · 2017-11-28

**Rating:** 6
**Confidence:** 4

**Review:**


The paper proposes a generative model for images that does no require to learn a discriminator (as in GAN’s) or learned embedding. The proposed generator is obtained by learning an inverse operator for a scattering transform.

The paper is well written and clear. The main contribution of the work is to show that one can design an embedding with some desirable properties and recover, to a good degree, most of the interesting aspects of generative models. However, the model doesn’t seem to be able to produce high quality samples. In my view, having a learned pseudo-inverse for scattering coefficients is interesting on its own right. The authors should show more clearly the generalization capabilities to test samples. Is the network able to invert images that follow the train distribution but are not in the training set?

As the authors point out, the representation is non-invertible. It seems that using an L2 loss in pixel space for training the generator would necessarily lead to blurred reconstructions (and samples) (as it produces a point estimate). Unless the generator overfits the training data, but then it would not generalize. The reason being that many images would lie in the level set for a given feature vector, and the generator cannot deterministically disambiguate which one to match.

The sampling method described in Section 3.2 does not suffer from this problem, although as the authors point out, a good initialization is required. Would it make sense to combine the two? Use the generator network to produce a good initial condition and then refine it with the iterative procedure.

This property is exploited in the conditional generation setting in:

Bruna, J. et al "Super-resolution with deep convolutional sufficient statistics." arXiv preprint arXiv:1511.05666 (2015).

The samples produced by the model are of poorer quality than those obtained with GAN’s. Clearly the model is assigning mass to regions of the space where there are not valid images.  (similar effect that suffer models train with MLE). Could you please comment on this point?

The title is a bit misleading in my view. “Analyzing GANs” suggests analyzing the model in general, this is, its architecture and training method (e.g. loss functions etc). However the analysis concentrates in the structure of the generator and the particular case of inverting scattering coefficients.

However, I do find very interesting the analysis provided in Section 3.2. The idea of using meaningful intermediate (and stable) targets for the first two layers seems like a very good idea. Are there any practical differences in terms of quality of the results? This might show in more complex datasets.

Could you please provide details on what is the dimensionality of the scattering representation at different scales? Say, how many coefficients are in S_5?

In Figure 3, it would be good to show some interpolation results for test images as well, to have a visual reference.

The authors mention that considering the network as a memory storage would allow to better recover known faces from unknown faces. It seems that it would be known from unknown images. Meaning, it is not clear why this method would generalize to novel image from the same individuals. Also, the memory would be quite rigid, as adding a new image would require adapting the generator.

Other minor points:

Last paragraph of page 1, “Th inverse \Phi…” is missing the ‘e’.

Some references (to figures or citations) seem to be missing, e.g. at the end of page 4, at the beginning of page 5, before equation (6).

Also, some citations should be corrected, for instance, at the end of the first paragraph of Section 3.1:

“… wavelet filters Malat (2016).”

Sould be:

“... wavelet filters (Malat, 2016).”

First paragraph of Section 3.3. The word generator is repeated.

---

> ### Author Response · Authors · 2017-12-28
> **Answers to AnonReviewer3**
>
> Remark: Please do not miss the thank you comment (above) for all the reviews.
>
> Question: "In my view, having a learned pseudo-inverse for scattering coefficients is interesting on its own right. The authors should show more clearly the generalization capabilities to test samples. Is the network able to invert images that follow the train distribution but are not in the training set?"
>
> We have reorganized the paper so that this very important generalization point appears immediately, in Section 2.1 of the new version. We have thus inverted two sections. Figure 4 shows that the network is indeed able to generalize in the sense that it can invert test images that follow the distributions but which are not in the training set. The precision of this recovery depends upon the image complexity, which is quantified by Table 1.
>
> Question: "As the authors point out, the representation is non-invertible. It seems that using an L2 loss in pixel space for training the generator would necessarily lead to blurred reconstructions (and samples) (as it produces a point estimate). Unless the generator overfits the training data, but then it would not generalize. The reason being that many images would lie in the level set for a given feature vector, and the generator cannot deterministically disambiguate which one to match."
>
> The blur is in fact not due to the non-invertibility of the embedding. We checked this by using an invertible embedding, by reducing the scattering scale $2^J$ from $2^5$ in the original paper to $2^4$ in this version. Figure 2(a,b) shows that this embedding operator is nearly invertible. However, as we now further emphasize in Section 2.1, the convolutional network does not invert exactly the embedding operator (here the scattering). It makes a regularized inversion on the training images. It is the regularization induced by the convolutional network structure which allows the generator to build random models of complex images. Figure 2(c,d) shows that it recovers much better images than what would have obtained by inverting the scattering embedding operator, this is also now explained in Section 2.1.
>
> The regularized inversion is based on some form of memorization of information in the training samples, which does not seem to be sufficient for complex images. The blur is indeed smaller for images of polygons. It may also be that the blur is partly due to instabilities in the optimization. We now explain this point in the experiments section.
>
> Question: "The sampling method described in Section 3.2 does not suffer from this problem, although as the authors point out, a good initialization is required. Would it make sense to combine the two? Use the generator network to produce a good initial condition and then refine it with the iterative procedure. This property is exploited in the conditional generation setting in: Bruna, J. et al "Super-resolution with deep convolutional sufficient statistics." arXiv preprint arXiv:1511.05666 (2015)."
>
> As previously mentioned the goal is not to invert the scattering transform because it does not build good image models, as shown by the Figure 2. If we incorporate iterations from the inverse scattering transform, it degrades the model because the convolutional network generator becomes an inverse scattering transform.
>
> Question: "The samples produced by the model are of poorer quality than those obtained with GAN’s. Clearly the model is assigning mass to regions of the space where there are not valid images.  (similar effect that suffer models train with MLE). Could you please comment on this point?"
>
> We have reduced this effect by choosing a smaller maximum scattering scale $2^J$ with $J = 4$ as opposed to $J = 5$. GANs suffer from a diversity issue which means that they sample a limited part of the space. As the reviewer says, it seems that we do the opposite, the model assigns mass to regions where the images are not valid. We conjecture that there is a trade-off between image quality and diversity, this trade-off comes from a limited memory capacity of the network. However, understanding the generalization and memory capabilities of these models remains an open question.

---

> > ### Author Response · Authors · 2017-12-28
> > **Continuation**
> >
> > Question: "The title is a bit misleading in my view. “Analyzing GANs” suggests analyzing the model in general, this is, its architecture and training method (e.g. loss functions etc). However the analysis concentrates in the structure of the generator and the particular case of inverting scattering coefficients."
> >
> > We fully agree with this point. We thus propose to change the title to "Generative networks as inverse problems with scattering transforms" to emphasize our inverse problem approach, which is indeed different from GANs. This is the title of the new version.
> >
> > Question: "However, I do find very interesting the analysis provided in Section 3.2. The idea of using meaningful intermediate (and stable) targets for the first two layers seems like a very good idea. Are there any practical differences in terms of quality of the results? This might show in more complex datasets."
> >
> > There were slight numerical differences but no visual difference in the quality of the training and test images. To address the previous point of the reviewer, in this new version, we reduced the scattering scale from $2^5$ to $2^4$, which also improved image qualities. We thus do not distinguish the invertibility from the non-invertibility range of the scattering, which also simplifies explanations. We are thus now using a single global architecture which is the same as the DCGAN generator (Radford et al. 2016). We are currently exploring better the idea of using meaningful intermediate targets along the generative network. However, these results extend the paper considerably; therefore, they will be better explained in future work.
> >
> > Question "Could you please provide details on what is the dimensionality of the scattering representation at different scales? Say, how many coefficients are in $S_5$?"
> >
> > We gave the ratio $\alpha_j$ between the number of image coefficients and the size of each layer $S_j$. For $j = 5$ then $\alpha_j = 0.66$ which means that $S_5$ has about twice less coefficients than the number of pixels in the image, which is $64^2$. If $j = 4$ then $\alpha_j = 1.63$ and $S_4(x)$ thus has more coefficients than the image $x$ which explains why it is invertible. We now give these numbers.
> >
> > Question: "In Figure 3, it would be good to show some interpolation results for test images as well, to have a visual reference."
> >
> > This is done in Figure 6 in the new version.
> >
> > Question: "The authors mention that considering the network as a memory storage would allow to better recover known faces from unknown faces. It seems that it would be known from unknown images. Meaning, it is not clear why this method would generalize to novel image from the same individuals. Also, the memory would be quite rigid, as adding a new image would require adapting the generator."
> >
> > The network has some form of memory since it recovers high dimensional images from lower dimensional input vectors. It can be considered as an associative memory in the sense that it is content addressable. From an image $x$ sampled from $X$, we can compute an address $z = \Sigma_d^{-1/2} (S_J (x) - \mu)$ from which we can reconstruct an approximation of $x$. This is what is usually called an associative memory (e.g., Hopfield networks). Indeed, it clearly depends upon the generalization capabilities of the network.
> >
> > The memory is indeed rigid in the sense that it requires modifying all coefficients to add a single image, but this is the case of any distributed associative memory such as Hopfield networks. We agree that the ability to add a new face easily in the network is key to have an effective memory.
> >
> > We have included all the minor points of the reviewer.

---

### Author Response · Authors · 2017-12-28
**Thanks to the reviewers**

We would like to thank very much the reviewers who helped us to understand the weak points in the paper writing. As a consequence of these remarks, we have changed many explanations and some elements of the organization of the paper, which hopefully will make things clearer. We apologize for the heavy modifications that resulted in the paper, but we felt that given the positive and important feedback of the reviewers,
we should improve the paper presentation, at the cost of some reorganization. In the following, we answer each reviewer's remarks and relate them to the paper modifications.

---

### Decision · Program_Chairs · 2018-01-29
**ICLR 2018 Conference Acceptance Decision**

**Decision:**

Accept (Poster)

**Comment:**

The paper got mixed scores of 4 (R1), 6 (R3), 8 (R2). R1 initially gave up after a few pages of reading, due to clarity problems. But looking over the revised version was much happier, so raised their score to 7. R2, who is knowledge about the area, was very positive about the paper, feeling it is a very interesting idea. R3 was also cautiously positive. The authors have absorbed the comments by the reviewers to make significant changes to the paper. The AC feels the idea is interesting, even if the experimental results aren't that compelling, so feels the paper can be accepted.